# Chronological Progress of Blockchain in Science, Technology, Engineering and Math (STEM): A Systematic Analysis for Emerging Future Directions

**Anton Dziatkovskii [1], Uladzimir Hryneuski [1], Alexandra Krylova [1] and Adrian Chun Minh Loy [1,2,*]**

1   Platinum Software Development Company, 67-170, Punane St., Lasnamae, Distri., 13619 Tallin, Estonia
2   Chemical Engineering Department, Monash University, Clayton, VIC 3180, Australia
*   Correspondence: adrian.loy@monash.edu

**Abstract:** The emergence of Industry 4.0 has awoken the adoption of blockchain as a key factor to enhance the industrial supply chain across the globe, enabling cost-effective and fast-paced delivery of products and services, ownership of products with privacy, and high security as well as traceability. This new digital horizon is underpinning the future direction of humankind, aligning with the Sustainable Development Goal themes of Good Health and Well-being (SDG3) and Sustainable Cities and Communities (SDG 11). Thus, the main objective of this paper is to elucidate the adoption of blockchain technology in Science, Technology, Engineering, and Math (STEM) disciplines by determining of the key academic research players and the evolution of blockchain in different fields. It begins by clarifying the definition of these concepts, followed by a discussion regarding the chronological progress of blockchain over time, an evaluation of the adoption of blockchain technologies in different key research areas, and lastly, providing comments on several directions to guide practitioners in developing a sustainable global blockchain roadmap in education science.

**Keywords:** STEM; blockchain; chronological analysis; science; technology

## 1. Introduction

The Industry 4.0 era, a paradigm revolution from a mechanical technology emphasis era to a digital era, encompasses the interconnectivity of both Internet of Things (IoT) and smart manufacturing [1,2]. Notably, Industry 4.0 offers a more comprehensive, interlinked, and holistic approach to manufacturing in terms of massive capabilities of storage and computing via a cloud servers, effective process handling, and optimization via machine learning, and better visualization in decision making via digital twin's technology [3–5]. Beyond the innovation bottlenecks imposed by Industrial 1.0 to 3.0, Web 3.0 is envisioned as an open and decentralized version of the internet which enhances the shifting process of conventional industrial automation into the form of Digital Twins Cyber-Physical systems [6,7]. On this basis, Web 3.0 can be considered as a key aspect in the revolution of Industrial 4.0 under the prism of social, economic, and cultural of human mankind, including (a) social interaction between users in a virtual ecosystem [8,9]; (b) cryptocurrency as a certain domain of economic value [10]; and (c) allowing private ownership of self-production [11].

On the other hand, unsurprisingly, blockchain technology has become the center of focus for the realization of the new digital industrial 4.0 era due to its preferences for trustworthiness, decentralization, and also tamper-proofing, coordinating the theme between Industrial 4.0 and Web 3.0. Blockchain is a distributed ledger technology where an array of an individual blocks of transactions are stored anonymously in a decentralized ecosystem in which the users are in full control of their transactions, without the necessity of a trusted source or third-party authorities to verify the information [12]. In the last

decade, blockchain has empowered and evolved to be a paradigm in the information technology perception, in which there are various broadly applied applications ranging from government to industries (i.e., crowdfunding [13,14], supply chain management [15,16], voting services [17], healthcare [18]. This is because blockchain technology has unwrapped ample knowledge gaps for the research community and provides a great deal of capacity in strengthening the cross-research development between different research areas [19,20]. With the combination of traits from a diverse range of areas such as cryptography, peer-to-peer networking, and decentralized ecosystem, research targeted on blockchain repeatedly surpasses the restrictions of the conventional research technique [21,22].

Furthermore, blockchain can be designated into a different form of network management, ranging from public to private sectors as well as the federal government [23,24]. Notably, integrity verification is one of the most important features of blockchain, which can save data anonymously and also link transactions for the creation of services (i.e., provenance and counterfeit, intellectual property (IP) management, and insurance). For instance, Ascribe (https://goascribe.com/, accessed on 5 February 2022) utilizes blockchain for linking digital content with creators, enabling a smooth transfer of ownership along with digital assets of loans without hassle in the process or high transfer service fees. Meanwhile, from a smart cities and societies perspective, blockchain can be a potential authority for exhilarating innovation by coupling Fog computing models with IoT blockchain [25]. One of the examples projected by Heidari and co-authors is that they reported that the digital taxonomy for the management of smart cities and societies could provide authority, privacy, and security to human life, enhancing the well-being of humans [26]. Besides aiding authority and privacy problem elimination, blockchain also enables the empowerment of new business models in the marketplace, which reduces the maintenance cost of data storing, enhances the security and privacy of ownership and improves the transaction's speed. Alas, the real-time adoption of blockchain is still in its infancy, specifically in science and engineering applications, lacking in-depth fundamental adoption that bridges academia, industry, and policymakers together.

The size and the composition of the Science, Technology, Engineering, and Math (STEM) workforce have gradually increased over the years, showing a positive trend in global demand, and experts have forecast that this number will continually rise at ~two- to three-fold in the next decades [27]. Recently, the government became aware that design-based learning principles, incorporating observation, prediction, communication, classification, and measurement skills, are essential skills for STEM disciplines [28]. Under this rule of thumb, the adoption of blockchain can be steep in a culture where "education" or "knowledge" or "information" sharing can be done easily and where everyone has a right to obtain it freely, without paying a high cost for access. One of the benefits is, blockchain would eliminate intermediaries' barriers between third parties, allowing for much more direct access to upskilling your knowledge and providing meaningful credit that leads to degree completion. For example, blockchain helps to capitalize on a number of STEM disciplines such as computing engineering and data science, which alleviate the shortage of workers in this niche area. Furthermore, blockchain could help in providing real-time STEM teaching where everyone can be placed in the same room, same thinking, and same ideology via metaverse technology, ensuring everyone is at the same pace of learning without being left out. On top of that, over the decades, there have been more advantages of the adoption of blockchain that have been reported in STEM fields, such as healthcare, data security management, finance, advanced manufacturing, and urban development, (see Figure 1).

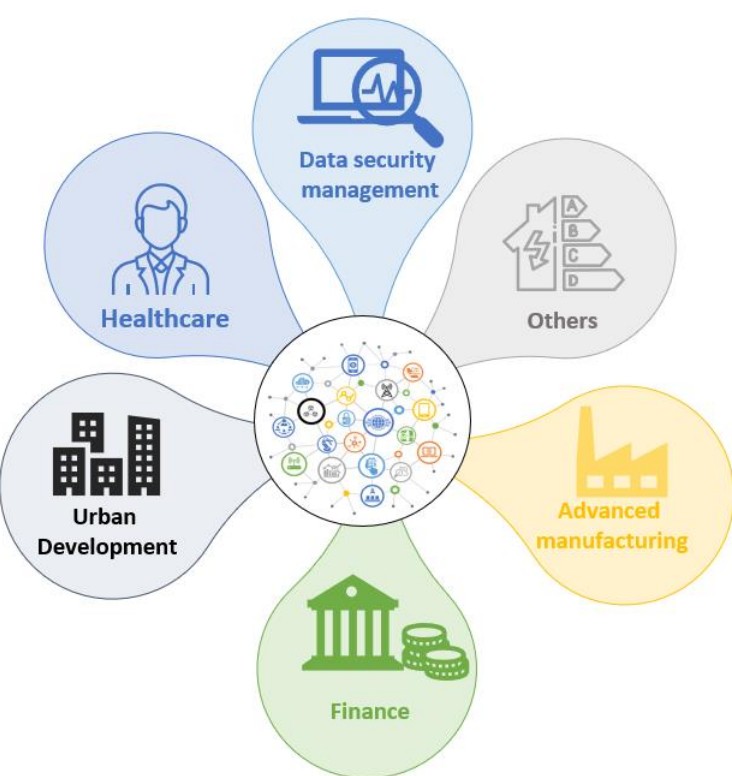

**Figure 1.** Blockchain adoption in various STEM applications.

To the best of the authors' knowledge, this is the first study that sheds spotlight on the (a) emergence of blockchain technology and Web 3.0, (b) the blockchain platform operation in STEM disciplines, (c) the key academic players working along blockchain technology, and (d) the impacts of blockchain on humans and the environment. By doing so, we have evaluated numerous features, including benefits, challenges, and affordability of adoption of blockchain in each STEM discipline's technology. We have also taken into consideration the future work in great detail, highlighting all of the issues that need to be addressed. Some research questions are formulated to cater to the research gap:

Q1. What is the maturity level of blockchain applications in the STEM discipline?

Q2. How could blockchain fog computing, machine learning, IoT and, IoV aid in Web 3.0?

Q3. How many publications related to blockchain application in STEM disciplines can be found in the literature?

Q4. Who are the key players in blockchain applications?

Q5. What are the benefits of the adoption of blockchain technology in STEM applications?

Q6. What are the limitations or challenges that exist in the adoption phase of blockchain in STEM disciplines?

## 2. Methods

Motivated by the aforementioned observations, this study aims to provide an insight into the research activities and dynamics related to blockchain in terms of:

1. The revolution of web technology from the beginning of the late 19th century until today.

2. The "hotspot" of blockchain application and the key academic players in STEM via bibliometric analysis.

3. The chronological progress of blockchain in the disciplines of STEM, not limited to economic, finance, energy, and chemical research areas.

Over the years, bibliometric analysis has been acknowledged as one of the most preferred techniques for analyzing research trends, narrowing and understanding the net-

work clustering—the relationship of the research theme. Most importantly, it can provide a big overview of the "hot topic" of the current research trend and the emerging state-of-art technology. The literature was retrieved for the last two decades (2000–2021) using the keyword search of "Blockchain" using the database of Scopus. Consequently, we utilize the maps generated by VOSviewer to carry out a comprehensive analysis of blockchain (https://www.vosviewer.com/, accessed on 5 February 2022).

To thoroughly cover the international research of "blockchain technology", we have utilized the Scopus@ database to acquire plentiful professional and scientific literature. Exported records from Scopus comprised complete and comprehensive information (cited references, full records exported to CSV files) on publication year, author, institution, and source journal. The search categories mostly focused on keywords, so that they discover correlation theories and research content in this context. In total, 46,164 valid documents were extracted from Scopus, with pre-analysis and comparison, keywords search for TITLE-ABS-KEY (blockchain AND technology); TITLE-ABS-KEY (blockchain AND engineering); TITLE-ABS-KEY (blockchain AND science); TITLE-ABS-KEY (blockchain AND mathematics); TITLE-ABS-KEY (blockchain AND mathematics); TITLE-ABS-KEY (blockchain AND business management); TITLE-ABS-KEY (blockchain AND energy); TITLE-ABS-KEY (blockchain AND engineering); TITLE-ABS-KEY (blockchain AND computer science); TITLE-ABS-KEY (blockchain AND environmental science); TITLE-ABS-KEY (blockchain AND decision science); TITLE-ABS-KEY (blockchain AND physics and astronomy); and the retrieval period was from 2016 to 2022, ensuring high reliability of the results.

## 3. Literature Review

The literature review is conducted to map the research systematically conducted in STEM disciplines and to identify any existing gaps in knowledge to which the scientific community can contribute.

### 3.1. Evaluation of the Evolutionary Progress of Web Technology

The spark of interest in Web technology can be backdated to the year 2004 (Web 2.0), when it capitulated the world from a static desktop web page to interactive experiences along with user-generated content (i.e., Airbnb, Friendster, and Facebook). This era was mainly driven by three fundamental layers, namely social, mobile, and cloud in limited and superficial conditions. Nonetheless, the Web 2.0 era is still fruitful, and currently, we are noticing the first glimpses of growth arising out of the next gigantic epitome shift in applications of the internet towards the Web 3.0 era.

Web 3.0 is a fresh new formation of architecture that fortifies the internet. The term was coined by Gavin Wood, and in the recent past has harvested substantial currency among specific futurists [29]. For a fair comparison purpose, Web 1.0 is represented through rudimentary read-only passivity in users, and those users browse a quite poorly organized framework at low bandwidths along with limited accessibility; meanwhile, Web 2.0 is represented through a dualist interaction between consumers and the producers of content. For instance, posting, blogging, and tweeting all portray active methods of content creation on these platforms. Accordingly, the Web 2.0 issue that was discussed by [30], stated that platforms on which all users interact are core pillars of power in their own right. The platforms such as Facebook (Meta) provide them with an astonishing degree of control over the structure of the internet. Correspondingly, in the streaming video category there are Netflix and YouTube; Twitter in microblogging; Amazon, eBay, and a few more for e-commerce; LinkedIn for professional networking; and lastly Google in the search engine segment. All of these will be curators of content and also ushers of useful information to the public. These oligopolies are denoted as the "Big Six" [31].

During the past few decades, the above-mentioned oligopolies have developed from, agile, low-profile outfits into powerhouses that govern their sub-areas, and also procured any firm that raised even the remotest challenge. Before adopting the Web 3.0 technology, the internet speed (5G) was bizarrely slower ($\times$10–100 times) than that of the internet speed

in the year 2012 (4G) and 1999 (3G), respectively. Nonetheless, the question is to envision an improved architecture, and not remain hindered in the current scheme through discussing apprehension of stability to it; mainly since numerous people do recollect an age before the internet exist [10]. Overall, Web 3.0 is postulated on a "user-centric" architecture with a prominent characteristic of decentralized blockchain protocol [32]. Web3 is a group of protocols to offer core components for application creators. These components replace the old-fashioned web technologies such as AJAX, MySQL, and HTTP, and offer a new way of making applications. The user is provided with robust and verifiable assurances regarding the information they are receiving, the information they are giving, what they are paying, and In return what they are receiving. So basically, users act for themselves in low-barrier markets, and there is a guarantee of censorship along with monopolization that has rarer places to cover up.

### 3.2. Evaluation of the Chronological Progress of Blockchain

Figure 2 illustrates the timeline of the revolution of blockchain technology. The first emergence of digital cryptocurrency, known as Bitcoin, was invented by [33]. Several researchers have contributed substantial concern to blockchain technology. Nakamoto theorized the first-ever blockchain, where the technology has developed and established its way into several applications beyond cryptocurrencies.

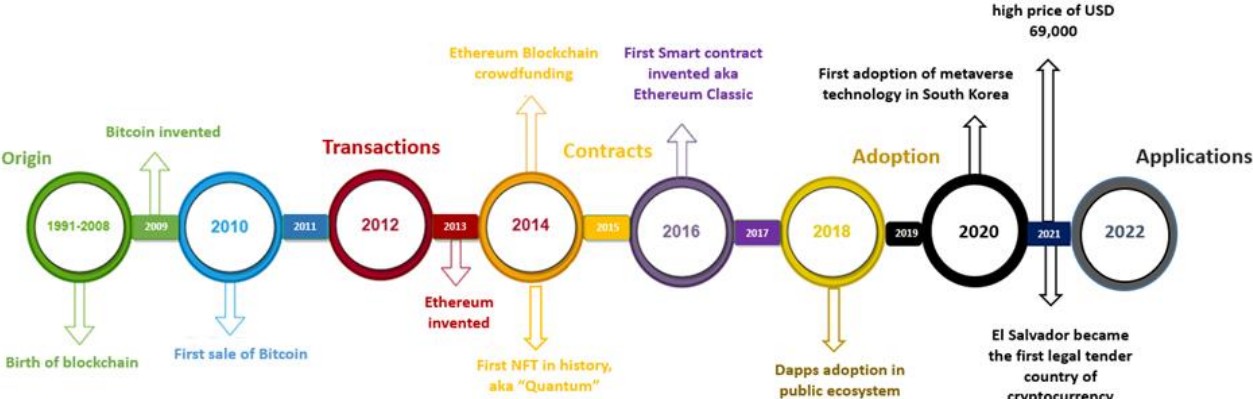

**Figure 2.** The revolution timeline of blockchain technology from 1991–2022.

Bitcoin sorts transactions and then clusters them into a controlled-sized structure known as blocks that share similar timestamps. The miners of the network (nodes) are accountable for connecting blocks in chronological order, and each block has the hash of an earlier block to help create a blockchain [34]. Consequently, the structure of blockchain achieves a robust registry of all transactions. Since Bitcoin, which is an application of blockchain, was introduced to the world, several applications have been gathered, all of them seeking to influence the capabilities of the digital ledger. As a result, blockchain history comprises a great number of applications that have developed with the evolution of the technology. The initial phase, which is considered the emergence of Bitcoin, is blockchain 1.0.

After that, concerned by Bitcoin's restrictions, Ethereum was introduced [35], which essentially delivers an infrastructure, similar to an operating system (OS), that can be built by everyone with their applications on top minus the necessity of high-priced enlargement of an own blockchain. Ethereum was invented as a novel public blockchain with additional perks related to Bitcoin, an expansion that ended up as a crucial period in blockchain history. Ethereum introduced a function called smart contracts (SCs) that are programmable in a certain type of languages, namely Java, GO, and Solidity [36]. This new feature of Ethereum provided a platform to create decentralized applications and progressed to become the largest application of Blockchain Terminal (A gateway that provides real-time valida-

tion, security and compliance for fund management) considering its aptitude to upkeep smart contracts utilized to execute numerous tasks. Furthermore, blockchain evolution does not rest solely on Ethereum and Bitcoin, but also other ecosystem such as Polkadot and Uniswap.

Amongst the most provocative things in cryptocurrency is how to resolve the transaction. There exist two leading types of consensus mechanisms, namely Proof of Work (PoW) and Proof of Stake (PoS). The PoW involves more electricity than PoS to PoW to authenticate nodes. PoS has a key issue in that it undergoes the likelihood of chain split, enabling process transactions quicker and at a lower cost than PoW, which is key for scalability (Cao et al., 2020). Recently, several other developments have cropped up to utilize the capabilities of blockchain technology, such as NEO cryptocurrency. It is a public smart contract cryptocurrency that portrays a PoS open-source that utilizes Delegated Byzantine Fault Tolerance (DBFT) technology; alternatively, it can support approximately 10,000 transactions per second. This consensus mechanism consumes the lowest electricity along with eradicating the chance of chain splitting [37].

In late 2019, Non-Fungible Token (NFT), a type of cryptocurrency that is driven by smart contracts of Ethereum has gathered astonishing attraction from both scientific and industrial communities [38]. The importance of NFTs cannot be overstated. NFTs are acknowledged as one of the futures of digital assets or technology that can be a reference standard for digital transactions. Unlike traditional currencies and tokens, NFTs are secure and decentralized. In other words, they cannot be controlled or stolen by any one entity. Also, NFTs are tamper-proof or immune to cyberattacks and cannot be altered or destroyed. This makes them a reliable way to store and exchange information.

Although NFTs have a marvelous influence on the present decentralized markets along with upcoming business prospects, the NFT technologies are however in quite a premature phase. Some possible challenges are mandatory to be sensibly embarked upon, whereas some favorable opportunities should be emphasized [39]. Furthermore, literature on NFTs, ranging from forum posts, codes, and blogs along with other foundations, are obtainable to the public. The first-ever NFT on Ethereum is CryptoPunks [40], whereas CryptoKitties (2021) place NFTs on notice with the gamification of the breeding mechanism. These participants fiercely contested at quite high prices for public sale the rare cats [41]. An imperative characteristic of NFTs is uniqueness, which makes NFTs appropriate for identity depiction, for instance, assets that are private and might be traded and transferred without restrictions.

NFTs are in a variety of interests these days, from play-to-earn gaming to self-collection. Lately, this technology has been adopted in STEM disciplines, such as Biotechnology, Education, and Engineering [42,43]. The first NFT in biotechnology is a breakthrough. It opens up a whole new world of possibilities for medical treatments and research. With this new technology, we can create customized treatments for patients with specific conditions. Additionally, we can use NFTs to create new models of disease and test potential treatments. This is a huge step forward for medicine and science. George Church and his co-founded company Nebula Genomics have advertised their plan to sell an NFT of Church's genome. Church is a geneticist at Harvard University in Cambridge, Massachusetts, who helped to launch the Human Genome Project, and is well known for controversial proposals, including resurrecting the woolly mammoth and creating a dating app based on DNA [38]. On the other hand, Jetking Infotrain Limited, a listed Indian computer networking organization, declares the launch of an assortment of 10,000 exceptional Web 3.0 Lion (NFTs) on the Ethereum blockchain. These NFTs will be accessible for exchanging enrolment and course expenses offered by the organization. In the metaverse, these sorts of virtual assets originated with certificates called NFTs that show ownership [44,45].

As a whole, blockchain technology is achieving new milestones day by day from cryptocurrency to NFTs, followed by the state-of-art metaverse. The metaverse is a virtual shared area that is open to everyone. It is a comprehensive term that states the entire digital as well as virtual world [46]. At present, numerous initiatives are emerging digital twins of

the physical world we live in plus access to this digital world through the network. The metaverse is expected to address the issues of physical infrastructure in offering similar education. Instead of purchasing costly educational objects, they can be presented in the digital world for only one percent of the actual cost using virtual reality equipment. This state-of-the-art technology will lessen the difference between urban and rural students as well as providing the same level of education to everyone, no matter in which part of the world [47]. For instance, the rise of state-of-the-art "Metaverse-Digital Twins" technology is a life-changing frontier, where health care providers can interact within the augmented world globally and can immerse themselves in clinical discussion or perform a real-time virtual cardiovascular surgery that gives real experiences, which could increase the success rate of the intervention while minimizing the associated operative complications and risks [48]. The avatar-based 3D virtual ecosystem enables an easier collaboration among the academic researchers and the practitioner, connecting the science to real-world practical applications. Also, the Internet of Vehicles (IoV), another paradigm for advanced automotive networks. With the aids of 5G, data sharing and storage from the autonomous vehicles can be implemented in real time, meaning better road safety and lower traffic causalities. Through the decentralized blockchain ecosystem, secure and privacy-preserving services for the vehicles' network can be achieved via the device-to-device (D2D) communication technology [49].

## 4. Discussion

### 4.1. Bibliometric—Hotspot Analysis

The publication's annual tendency encourages discovering the development stage, knowledge accretion, and blockchain maturity. As illustrated in Figure 3, the total number of articles published in 2016 was merely 85. In 2017, articles associated with blockchain showed rather slow growth, just 463 articles, and 2018 was the start of massive growth in articles with 2206 articles. From 2019 onwards, the popularity of blockchain technology increased and 5858 articles were published, in 2020 reaching 10,509 articles. Further, 17,625 articles were published in 2021, which was the highest number of articles in a single year, and 9409 articles in 2022.

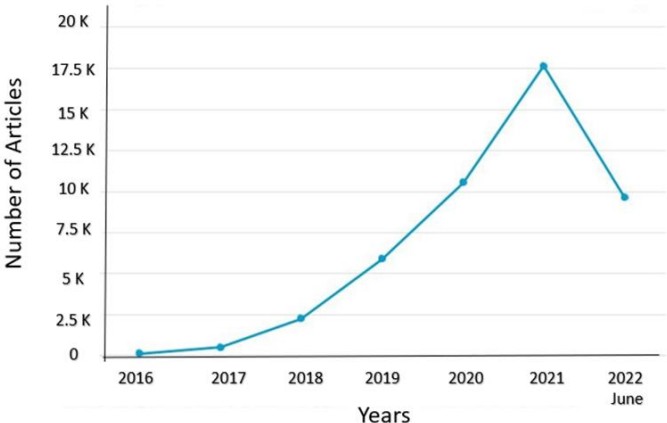

**Figure 3.** Publications per year based on keywords.

### 4.2. Subject Categorical Analysis

Blockchain literature in Scopus comprised several subject categories. The top 10 subject categories are illustrated in Figure 4, including computer science (31.2%) and engineering 21.4% as two of the hottest subject areas among other top subject areas based on keyword searches in the last decade. The number of publications in each group replicated the progress drifts of blockchain research in diverse areas. Generally, in the blockchain technology domain, a substantial proportion of current blockchain-related literature was mostly represented by cryptocurrencies (categorized under the field of finance). Nevertheless, based on our bibliometric analysis, the comparatively huge number of a diverse range

of applications also highlights the interdisciplinary potential of blockchain technology. The subcategories below deliver a sound grouping of the existing blockchain-enabled STEM applications based on scrutiny of available literature.

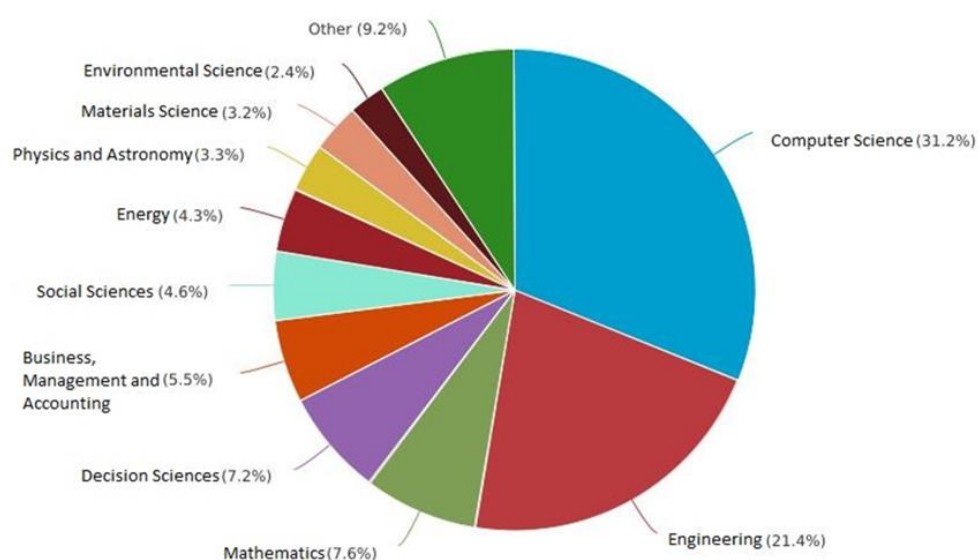

**Figure 4.** Subject categories analysis based on publications regarding the keywords.

### 4.2.1. Urban Development with IoT

Recently, IoT is expressively accelerating in the Information and Communication Technology (ICT) field [50–52]. IoT frameworks implement the centralized server-client model, assimilating things with cloud servers via the internet and therefore offering users several services [25]. IoT technologies along with blockchain are vast already through their expansion potential. Apart from that, these two fields are innumerably more entwined. For example, regardless of the downsides experienced by Wireless Sensor Networks (WSNs), which are a mainstay of scientific and human evolution [53], strong architectures of blockchains might be able to boost their IoT architecture by exploiting its ability while minimalizing insufficiencies [54]. Furthermore, growing consideration and investments for employing decentralized IoT systems are primarily impelled through blockchain technology and its intrinsic proficiencies [55].

Additionally, working in an automated and decentralized fashion empowers high scalability of the network along with effective management [56,57]. The interoperability nature of blockchain empowers independent and secure real-time payment services, amplifying traditional commerce and e-commerce along with public and private transportation systems [55,58,59]. Likewise, the employment of blockchain in ICT is highly mature; on this basis, the paradigm of blockchain technology in ICT can be expressed as "Urban Development for the creation of Future Smart Cities" [54,60]. For instance, Pazaitis and co-authors define an intellectual model of economics, which is blockchain-based decentralized cooperation, that may better aid the progress of social sharing between communities [61], such as infrastructure sharing and facilities sharing, and this will create never-before-seen efficiencies in city functions with a low environmental footprint. Sun et al. discuss the influence of developing blockchain technologies on three main aspects of the sharing economy (i.e., organization, technology, and human). They further examine in what way blockchain-based sharing services impart to smart cities [62], categorizing human perspective from the angle of a sharing service, underpinning the role of key infrastructure, human population, and education in urban development [63].

Henceforward, IoT and Smart Learning Environments are the core elements that need to be adopted in STEM education, especially Smart Urbanization Education, where the development of a digital ecosystem and deployment of IoT technologies can be centered, equipping everyone with high-quality education (eliminating poverty) and benefiting the social economy of the developing or rural countries (aligned with SDGs goal 1, 4, and 11).

4.2.2. Finance

On the other hand, blockchain technology is also widely employed in several financial-related fields, namely, business services, financial assets settlement, economic transactions along with market prediction. Blockchain is likely to represent a crucial part of the sustainable growth of the global economy, as a result, favoring all of the consumers around the globe alongside the existing banking system [63]. Blockchain technology provides a considerable revolution to capital markets along with an efficient method of performing derivatives transactions.

Furthermore, if we associate the traditional fiat currencies with cryptocurrencies, which have an indigent reserve of values with no government interference and therefore consist of reduced-price constancy and also offer a swift method of the cheap medium of exchange. However, the value of cryptocurrencies is still measured in fiat currency. Another potential area is Prediction Marketplace Systems (PMS). The PMS aids as oracles might influence cryptocurrencies and businesses. The peer-to-peer networking (P2P) implementations of blockchain-based PMS, a PoW category that permits a swifter transaction than Bitcoin, an open-source cryptocurrency featuring Scrypt Merged mining that permits the user(s) to trade shares before an event happens under the paradigm of the wisdom of the crowds [64].

Similarly, several other fields are financial-oriented, namely syndicated loans, contingent convertible bonds, commercial property, proxy voting, automated compliance, over-the-counter market, and asset rehypothecation [65]. Over the years, blockchain implementation in the financial area has ultimately aided in cost savings in fields, namely centralized operations, compliance, business operations, and central finance reporting. On this basis, blockchain perhaps is an important foundation for stimulating invention through automated, enhanced, and improved business processes [66]. There are several e-business models evolving and these models rely upon the IoT blockchain. For example, Zhang and Wen presented a smart contracts-based business model in which a blockchain distributed database is utilized to accomplish transactions amongst different devices [56]. Additionally, blockchain applications deliver commercialization prospects and substantial performance improvement, empowering IoT firms to revamp their operations along with strengthening integrity in the e-commerce domain. Simultaneously, the applications based on blockchain technology could benefit enterprises through implementing them as business process management systems. On such a basis, blockchain can play a vital role in maintaining each business process, and smart contracts may be engaged in executing the business routing, thus shrinking cost, coupled with streamlining intra-organizational procedures.

Furthermore, in the supply chain domain, blockchain technology plays the most crucial role and is most likely to provide an upsurge in accountability along with transparency. After primarily sourced merchandise is mass-produced and circulated to consumers, goods are determined as a supply chain. Therefore, supply chain managers conclusively intend to produce efficient goods for supply and ensure end-user satisfaction despite having to waver in the budget. By utilizing blockchain technology, logistic firms can have product tracking data and sustainability along with enhanced quality products possibly distributed within the entire supply chain network, which will help in enhancing time, risk management, and cost as well. Additionally, in the supply chain, the utilization of blockchain-based applications can preserve security, resulting in robust contract management systems between third- and fourth-party logistics [67]. With the help of a smart contract, payment can be processed automatically during the process of returning a product to the supplier or issuer. The payment can be automated with a smart contract when returning a product

to the issuer or seller. The supply chain can have consensus-verified real-time tracking, connecting all members on the same platform.

As a whole, a step beyond the disruption and gaining the knowledge and skills to tackle financial services innovation via blockchain can be obtained in Fintech education. Within the Fintech education boundaries, students can experience the digital real-life challenges that reflect the dynamic nature of the current financial landscape and the transition from the traditional to the new digital era (aligned with SDGs 4, 8, and 12).

### 4.2.3. Healthcare Applications

Blockchain technology has progressed and is able to play a crucial role in the healthcare sector through numerous applications in fields, namely, longitudinal healthcare records, public healthcare management, user-oriented medical research, online patient access, patient medical data sharing, automated health claims settlement, clinical trial, drug counterfeiting, and precision medicine [68–70]. Particularly, blockchain technology along with the utilization of smart contracts could resolve issues of scientific reliability of finding things such as endpoint switching, data dredging, selective publication, and missing data in clinical trials.

One of the most promising adoptions of blockchain in healthcare is the integration of the blockchain with Electronic Healthcare Records (EHRs) of patients [71,72]. Basically, the patient's medical records, predictions, and data along with information relating to the condition of the patient and clinical development during the time of treatment are stored in EHR. So, by adopting such a technique, EHR based on a blockchain framework could be fruitful for patients to access and maintain the data that concomitantly promise privacy and security of the health records of the particular patient [73,74]. The EHR with of a blockchain-based system has multiple benefits: a distributed way of storing records that are public and also effortlessly verifiable over non-associated provider firms, no centralized holder or center for a hacker to breach the data or corrupt it, and data is updated and always obtainable while data from dissimilar sources is carried together in a single and amalgamated data repository [70].

In short, augmented reality and metaverse blockchain-related research are vital to be adopted in conventional healthcare management education as a sub-subject, i.e., digital health technology, augmented health infrastructure, 5G in healthcare. Healthcare practitioners do believe that the adoption of blockchain could upgrade the lifestyle of the patients, improve the healthcare quality and enhance the well-being of human mankind (aligned with SDGs 3, 4, and 17).

### 4.2.4. Advanced Manufacturing

The amalgamation of both virtual and physical structures in real-time data acquisition with blockchain-based digital twins (BBDTs) permitted live communications and permitted the operational effectiveness of unconventional manufacturing. In both biological and chemical preparations, the progress of bioassays frequently progresses in a composite and slow method to preserve high accuracy and precision [75]. As soon as they complete the authentication and optimization, the assay stays in use for a long period of time to decrease the most crucial factors, namely cost and waste. Consequently, in bulk production, the notion of lean manufacturing is quite fruitful to influence both capital expenditure and revenue [76]. For instance, BBDTs along with bioprocessing, together with multistage chemical production or else data acquisition and validation, biological cell cultivation, computational biomathematical optimization, and real-time data monitoring of bioreactors, perhaps steer to advanced operational efficiency along with a consistent product supply [77].

Furthermore, it provides participation of the inventor in tracking the traceability information of the supply chain of the dynamic manufacturing. This could provide a higher security than that of the traditional technologies because all parties must reach a consensus to place security blocks on top of encryption. It will reduce the chances of the data being stolen by another party [78]. Generally, the advanced manufacturing

engineering education would not be limited to manufacturing itself but does cover a broad range of areas, including artificial intelligence, additive manufacturing digital systems, and control, creating a step closer to the Industrial 4.0 roadmap (aligned with SDG 9 and 12).

### 4.2.5. Data Storage and Security Management

Primarily, this is a real test when there is a huge amount of data to be stored, operated, and gathered, but lately, the dawn of data mining and machine learning methods have been established to overcome conventional data storage management [79,80]. The applications and implementations based on blockchain technology are not merely improved data storage and security management but also helped by default suitability because all of the operations are verifiable [81].

Numerous challenges regarding privacy, security, and also centralized trust even now affect the advancement of Big Data in IoT. To manage the decentralization of distributed data processing, blockchain was being implemented. Contrasting to other procedures, blockchain permits data security and resolves the privacy problems in the Big Data area [82]. With the aim to stress, the significance of trustiness in the Big Data Management Field presented a trustworthy Big Data blockchain-based sharing model, which achieved to guarantee the secure flow of data resources along with integrating a smart contract technology. Yang et al. utilized a keyword search facility and cryptographic primitives to help create a framework allowing secure and distributed client data management [83]. In addition, read and search authorizations of the data can be approved through their holder to third parties.

Moreover, in the instance of secure data distribution, utilization of a blockchain-based solution for metadata supportive key functions also confers its involvement towards both sustainability and management of digital archives. A blockchain-based system allows better security in the marketplace of data trading. To individually send protected, consistent data to centralized cloud systems, fog computers must be suitably situated in fog [25]. Nevertheless, noteworthy problems might impede the cloud system because of two main downsides such as shutdown and reprocessing of fog computers. Foremost security characteristics are emphasized through blockchain technology due to digital signature along with consensus amongst fog computers that permit sharing and monitoring the authentic transactions.

As a whole, blockchain plays a very important part in information and communication technology education, providing ways to tackle sustainable management issues related to data security and trusts, (i.e., planning and design, bidding, obtaining consensus in secret documents, and preventing malware attacks. The anonymous feature of blockchain also acknowledged as most cost-effective approach to achieve a sustainable growth of the IoV network's ecosystem by keeping vehicles' identities hidden while maintaining their privacy (digital automobile world) [84]. The futuristic features of this technology is highly required in many big ICT companies such as Meta, Amazon, and Google, providing a huge job opportunity (aligned with SDGs 2, 16, and 17).

### 4.2.6. Other STEM Applications

The progress in nanotechnology together with both technologies, namely computational and communications, has built a way for unproblematic integration of electronics and also flexible sensors, which is proved to be quite valuable growth, exclusively in the health industry, as already pointed out. The IoT coupled with nanotechnology has built a worthy prospect for plentiful applications to be industrialized in health monitoring systems, agriculture, and several other applications. The progress of nanomachines consents nanodevices to create, compute, transmit, and gather data at the nanoscale. The communication of the linked devices at the nanoscale coupled with current classical communication networks at high speed has directed the development of the internet of nano things (IoNT). The IoNT benefits several sectors are found in agriculture, health monitoring systems, oil and gas, and multimedia coupled with several other applications [85].

One of the biggest achievements of the application is combining nanotechnology and blockchain for COVID-19 immunity passports [85]. This could ensure the authenticity of the health data of each individual, allowing policymakers and health agencies to formulate new plans and safety measures to contain the spread of COVID-19 and other diseases in a short period of time. Also, the integration of blockchain can degrade the COVID-19 pandemic by a variety of methods, including patient monitoring, management, imaging methods, and medication [86,87]. On the other hand, from the perspective of nano energy, blockchain is likely to resolve issues, namely energy storage, fuel, power generation, and solar cells. This will establish a further business opening in the advancement of energy storage technologies and renewable energy, aligning with the theme of Industrial 4.0 [86]. For instance, the adoption of blockchain for decentralized and distributed energy markets could facilitate the market mechanism to deliver the energy system of the future without government subsidy.

In short, the adoption of "Blockchain Energy & Utilities" into renewable energy engineering education is somehow highly beneficial, preparing the students in higher education and equipping them with the knowledge before the energy revolution (the blooming of the distributed energy resources via solar and electric car). On the other hand, blockchain education should also be encompassed in other education sectors such as agriculture (Agritech 5.0 revolution), nanotechnology (NanoWorld revolution), and pharmaceutical (digital pharma) (aligned with SDGs 6, 7, and 13)

### 4.3. Countries and Institutions Analysis

By examining the geographic and spatial distribution of articles from each country, we could know where the "hotspot" of blockchain application in STEM disciplines is. Based on Figure 5, a significant difference in the number of publications among various countries can be seen: China has the largest output of academic papers, which is around 14,473 articles, followed by India with 7513 and the USA with 5200 articles, respectively. In the current context, the articles published in these three countries are accommodating more than 70% of the total publication in this field. The country collaboration system of blockchain research is illustrated in Figure 6, and the size of nodes signified a diverse range of articles published related to the keywords of blockchain on a global scale; the bigger the size of the nodes means more related articles were published in that respective country. It is worth specifying that high significance hinted at the standing of nodes.

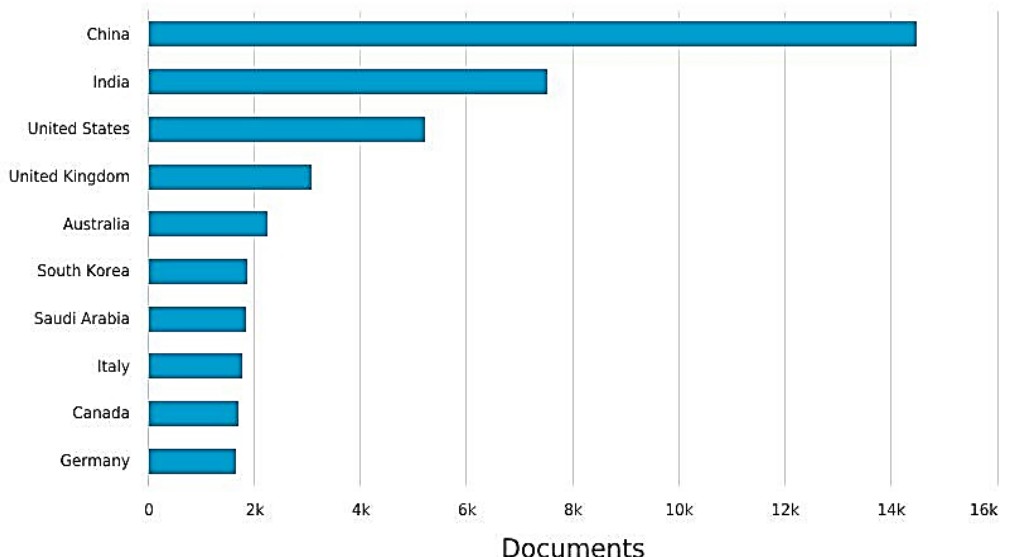

**Figure 5.** Number of STEM-blockchain-related articles published in various countries.

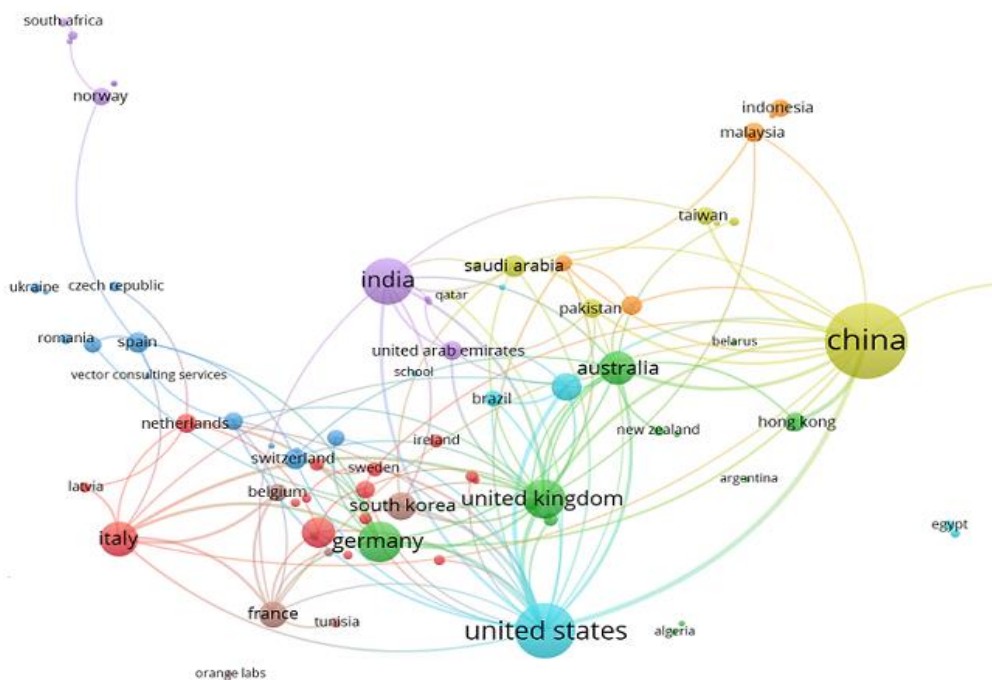

**Figure 6.** The visualization map of countries participating in blockchain research.

Figure 7 represents the top 10 institutions that contributed the most toward blockchain-STEM-related studies in the form of both collaborative and non-collaborative studies. Surprisingly, USA and Indian institutions are not listed in the top 10 ranking of the highest publication in Blockchain-STEM disciplines by the institution. The authors/researchers working in this field are concentrated in Asia, specifically in China, followed by Italy and Australia. Collectively, Peking University, Tsinghua University, and the UNSW have more active researchers in collaborative works, and the core research direction is found to be science and technology.

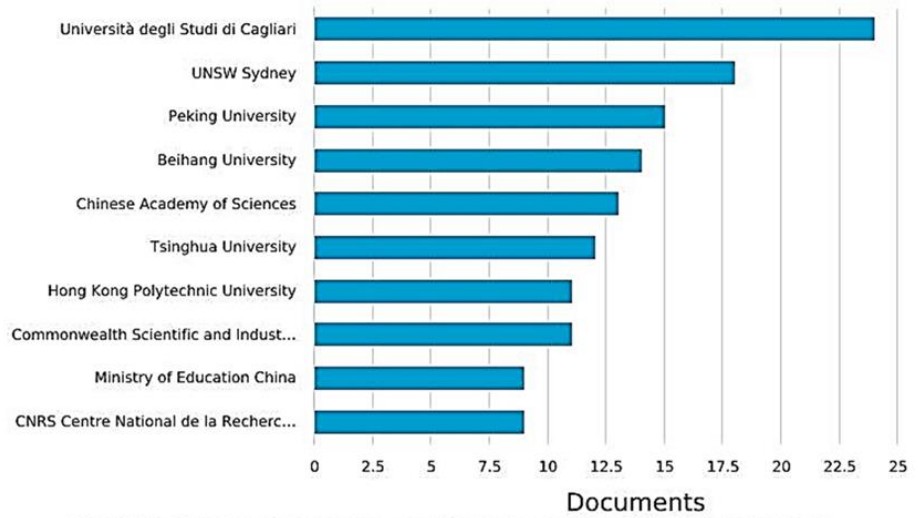

**Figure 7.** Top 10 institutions that published most STEM-blockchain-related work in literature.

## 5. Conclusions

In regard to implications for research on blockchain development, the growing number of publications on this topic indicates that there is a huge potential for blockchain that can be utilized in the field of STEM, shaping a sustainable digital future. Web 3.0, which includes Cloud, IoT, IoV, fog computing, and blockchain are the cutting-edge developments that

have enormous growth in STEM disciplines, and the integration of all these technologies will have a great positive influence on human well-being, communities, and nations in particular. Significantly, under the domain of blockchain, numerous research projects have projected good results and thus far have proven that it is a value-added criterion for research design and planning, especially in the field of science and technology. To provide a thorough understanding of the implication of blockchain, future works are suggested as below:

(a) Integrated Economy, Environmental, and Energy (3Es) analyses should be carried out to evaluate the sustainable metric of blockchain adoption in various disciplines.

(b) Strength, Weakness, Opportunity, and Threat (SWOT) analysis should be carried out to evaluate the possible challenges and opportunities across the field.

(c) Policy-tree decision study should be carried out to provide a preliminary overview of blockchain, enabling the decisionmakers to analyze and plan the roadmap of Industrial 4.0.

(d) Integrating renewable energy with blockchain technology is a new breakthrough that should be looked into.

(e) Combining blockchain technology with engineering research will be highly beneficial in terms of optimization, cost-effectiveness, and time-saving, specifically in the field of nanotechnology and biology.

Some limitations that we encountered are also enlightened as follows: The bibliometric analysis is limited to the database of WOS and Scopus where non-English blockchain related-documents or non-indexed literature are not being taken into consideration. Another shortcoming of this work is the literature that utilizes partially blockchain technology in their work, which lacks detailed descriptions of their methods or approaches that are being neglected (due to low accessibility and maturity). Nevertheless, as a whole, the findings in this paper still can serve as a preliminary practical guideline for the future development of STEM disciplines, not solely on the research on system design and data management but also for the realization of Industrial 4.0. Lastly, all the research questions that were derived at the beginning of the work are successfully addressed as follows:

A1. The maturity level of blockchain applications in STEM is still in the infant stage, and still has room for improvement in the near future.

A2. The adaption of fog computing, machine learning, IoT, and IoV are found to be highly beneficial in Web 3.0, speeding up the global digitalization and industrial 4.0 realization.

A3. The publications related to blockchain application in STEM disciplines are in the range of 10,000–50,000 pieces.

A4. Most academia key players that work in this field are from Canada, Australia, and the USA.

A5. The adoption of blockchain technology in STEM applications could further aid a blueprint for "peace and prosperity for people and the planet", aligned with the SDGs.

A6. The main challenges in the adoption phase of blockchain in STEM disciplines are found to be scalability, regulations, and cost of implementation.

**Author Contributions:** A.D.—Conceptualization; Methodology, writing—original draft preparation; U.H.—software, validation, formal analysis; A.K.—investigation, resources, data curation, A.C.M.L.—writing—original draft preparation, writing—review and editing, visualization, supervision, project administration, funding acquisition. All authors have read and agreed to the published version of the manuscript.

**Funding:** This research was funded by Australia Government through the commonwealth research training program.

**Institutional Review Board Statement:** Not applicable.

**Informed Consent Statement:** Not applicable.

**Data Availability Statement:** Not applicable.

**Conflicts of Interest:** The authors declare no conflict of interest.

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
