# Peer review of "Chronological Progress of Blockchain in Science, Technology, Engineering and Math (STEM): A Systematic Analysis for Emerging Future Directions"

_sustainability, doi:10.3390/su141912074_

Round 1

Reviewer 1 Report

 Here are my thoughts:

1- The paper's weak organization makes it difficult to follow.

2- Background and gap are missing from the abstract section.

3- The English language must be improved.

4- The introduction hierarchy is missing. The context, gap, how, why, and outcomes were all overlooked.

5- The contribution of the study should be clearly stated in the abstract part and in the introduction section with bullet points.

6- In addition, the introduction section lacks the paper's structure.

7- Please do not leave any sections blank. Section 3 should be filled out with a few sentences.

8-The important references are absent from the manuscript. Use and cite these works. Discuss these subjects in relation to the paper's topic:

https://www.sciencedirect.com/science/article/abs/pii/S2210670722004061 https://link.springer.com/article/10.1007/s00521-022-07424-w

9-The conclusion section is insufficient. It is necessary to mention the paper's limitations and flaws.

10-The paper lacks coherence and integration. It's a little chaotic.

Reviewer 2 Report

Authors have discussed and presented evolution of Blockchain in the field of Science, Technology, Engineering and Math systematically. The major observations during the review are as follows:

1. On Page 5, line no. 204 the term BCT is used first time  as abbreviation. Authors are suggested to write full form and explain a little bit.

2. On Page 7, Fig. 2 authors are suggested to indexing of articles i.e. articles belongs to Scopus Database, SCI database or others. Although it is mentioned in Fig. 3 as Scopus.

3. References 64 and 65 are not citied in running text.

4. On Page 9, line no. 367 the term P2P is used first time  as abbreviation. Authors are suggested to write full form.

5. On Page 10, line no. 431 authors used HER in place of EHR. Authors are suggested to correct it.

6. On Page 10, line no. 440, authors are suggested to use 5G in place of 5g.

7. On Page 23, Fig. 4 authors are suggested to indexing of articles i.e. articles belongs to Scopus Database, SCI database or others.

8. References are dually numbered.

At last authors are also suggested to provide solutions to different application areas.

Reviewer 3 Report

1.       Abstract of the manuscript is not clear, authors are talking about the bridging the gap. What kind of gaps must be specified. Authors also specifying mentioning “not limiting it to the development of web technology” but blockchain is already spread to the banking, gaming and metaverse so it would be not good to mention such comments in the abstract.

2.       Introduction of the paper is not clear, authors are describing the blockchain usages in different field, it must also specify the carried-out work in the manuscript. Objective is specified in the introduction but carried out work is not clear.

3.       A figure related to the blockchain and associated technologies covered in the manuscript can be added to the article introduction section.

4.       In the “method” section authors is telling about the retrieved paper, how these papers are retrieved?

Were there any tools used for that, if yes, its specifications set by the authors?

What was the keywords selected for search?

How authors identified their implementation area based on the mentioned “STEM”?

5.       There is need of the Literature review or carried out work to be added in the manuscript as the separate section before having discussion on the results.

6.       Section 3.1 contains good information, authors are suggested to create technology wise further subsection in 3.1 to increase the readability.

7.       A further categorization of figure 2 should be done, like block chain today cover different areas so further impact wise publication with blockchain in different fields.

8.       Conclusion should be more extensive, clearly describing the technology wise implications of the blockchain.

9.       Some more recent and technology wise references are required to be added in the article, as it looks some fields remain untouched in the study. Below are some of the links of the important articles that authors can consider to add

https://www.mdpi.com/1424-8220/22/14/5119
https://onlinelibrary.wiley.com/doi/abs/10.1002/ett.4520

Round 2

Reviewer 1 Report

It can be accepted.

Author Response

Thanks for your comment

Reviewer 3 Report

Authors have been made all the changes as suggested. Hence paper is accepted in its present form.

Author Response

Thanks for the comment